# Genome-Wide Identification of the Pectate Lyase Gene Family in Potato and Expression Analysis under Salt Stress

**DOI:** 10.3390/plants13101322

**Published:** 2024-05-11

**Authors:** Zhiqi Wang, Tao Liu, Wenbo Wu, Wenting Shi, Jian Shi, Fengyan Mo, Chong Du, Chaonan Wang, Zhongmin Yang

**Affiliations:** College of Horticulture, Xinjiang Agricultural University, Urumqi 830000, China; wangzhiqi1026@163.com (Z.W.); 13565545234@163.com (T.L.); wwb173522984@163.com (W.W.); shiwenting3800@126.com (W.S.); 17261163078@163.com (J.S.); 15086949885@163.com (F.M.); godv2018@163.com (C.D.); wcn0107@126.com (C.W.)

**Keywords:** potato, pectate lyase gene family, salt stress

## Abstract

Pectin is a structural polysaccharide and a major component of plant cell walls. Pectate lyases are a class of enzymes that degrade demethylated pectin by cleaving the α-1,4-glycosidic bond, and they play an important role in plant growth and development. Currently, little is known about the *PL* gene family members and their involvement in salt stress in potato. In this study, we utilized bioinformatics to identify members of the potato pectate lyase gene family and analyzed their gene and amino acid sequence characteristics. The results showed that a total of 27 members of the pectate lyase gene family were identified in potato. Phylogenetic tree analysis revealed that these genes were divided into eight groups. Analysis of their promoters indicated that several members’ promoter regions contained a significant number of hormone and stress response elements. Further, we found that several members responded positively to salt treatment under single salt and mixed salt stress. Since *StPL18* exhibited a consistent expression pattern under both single and mixed salt stress conditions, its subcellular localization was determined. The results indicated that StPL18 is localized in the endoplasmic reticulum membrane. The results will establish a foundation for analyzing the functions of potato pectate lyase family members and their expression under salt stress.

## 1. Introduction

Potato (*Solanum tuberosum* L.) is an important food crop in the world, second only to rice and wheat [1]. Globally, approximately 1.3 billion people consume potato as a staple food. Potato tubers contain a wide range of nutrients and are rich in potassium, iron, zinc, calcium, magnesium, and other mineral elements. The vitamin C content of potato is higher than that of wheat and other major food crops, and the protein and sugar content is higher than that of many vegetables [2]. These characteristics have made the potato become the only staple crop that can be grown in many parts of the world. In Africa, the expansion of potato cultivation has enhanced food security, prompting the United Nations to declare 2008 as the International Year of the Potato [3].

During the growth and development of the potato, the organ grows from small to large, and the plant grows from low to high. The intrinsic structural changes mainly involve an increase in the size of the cells and an increase in the number of cells [4]. In this process, the cell wall undergoes a process of disintegration and remodeling. The cell wall not only supports and protects plant cells, but it is also the first line of defense against adversity in plants [5]. The plant cell wall is composed of pectin, hemicellulose, and cellulose. They do not function independently but are interlinked to form a highly complex and dynamic network, which collectively determine the mechanical properties of the cell wall [6].

Pectin serves as a component of the primary cell wall of higher plant cells and maintains the mechanical strength of the cell wall. It consists of homogalacturonan (HG), xylogalacturonan (XAG), rhamnogalacturonan I (RG-I), and rhamnogalacturonan II (RG-II), which are arranged into “hairy regions” and “smooth regions”. The “hairy region” and “smooth region” are composed of rhamnogalacturonan I and rhamnogalacturonan II [7,8]. The degradation process requires the participation of polygalacturonase (PG), pectin methylesterase (PME), and pectate lyase (PL, PEL, or pectate lyase-like, PLL) [9].

Pectate lyase, also known as pectate transferase, is an enzyme that degrades the glycosidic bond of α-1,4-polygalacturonic acid residues in the smooth region of pectin through a β-elimination mechanism. This process results in unsaturated galacturonic acid residues at its non-reducing ends [9]. It was first discovered in *Erwinia carotovora* and *Bacillus polymyxa* in 1962 [10]. These pathogenic microorganisms can degrade the cell wall in the presence of calcium ions by secreting a variety of cell wall-degrading enzymes (CWDEs) [11], including pectate lyase. This process allows them to enter the cell, infect the tissue, and ultimately lead to diseases such as soft rot [12].

In higher plants, the pectate lyase gene was first isolated from tomato pollen and anthers [13]. Through conservative domain analysis, it was revealed that all plant PL proteins contain a Pec_Lyase_C domain, and some PL proteins also contain a Pec_lyase_N domain [14,15,16,17]. To date, *PL* family genes have been identified in multiple plants. For example, 26 *AtPLL* genes have been identified in *Arabidopsis thaliana* [16], 12 *OsPLL* genes in rice [15], 22 *SlPL* genes in tomato [18], and 46 *BrPLL* genes have been discovered in turnip [17].

Research has found that the expression levels of the 26 *AtPLL* genes in *A. thaliana* vary significantly in different parts of the plant. In seedlings treated with hormones, abiotic stress, and defense response inducers, the expression of some *PL* genes has changed significantly [19]. Cell wall composition is closely linked to a number of pectate lyase genes, and mutations in *PMR5* and *PMR6* (*AtPLL13*) cause alterations in the cell wall components of *A. thaliana*, resulting in increased pectin content. This modification enables *A. thaliana* to enhance its resistance to powdery mildew [20,21]. Among the 12 *OSPLL* genes in rice, *OsPLL1*, *OsPLL3*, *OsPLL4*, and *OsPLL12* exhibit higher expression levels in seedlings and young spikes, which may be associated with inflorescence development. The reduced expression of *OsPLL3* and *OsPLL4* results in male flower sterility [15].

The *SlPL* expression was significantly up-regulated during tomato fruit softening and was also induced by hormones such as 1-aminocyclopropane-1-carboxylic acid (ACC), abscisic acid (ABA), indole-3-acetic acid (IAA), and ethylene (ETH). However, *SlPL*, *SlPL5*, *SlPL6*, and *SlPL19* were downregulated in expression during fruit ripening. Silencing of *SlPL* increased the content of cellulose, hemicellulose, superoxide dismutase, peroxidase, and catalase in the fruit, and decreased water-soluble pectin and total pectin content [18]. At the same time, it has been shown that inhibiting *PL* gene expression in tomato can enhance fruit firmness and prolong shelf life without compromising the flavor, texture, and nutritional value of the fruit [22]. Currently, there are fewer studies on the *PL* gene family members in potato compared to other plants, both domestically and internationally. Therefore, this present study utilizes bioinformatics analysis to explore the *StPL* family genes in the potato genome. It conducts an analysis of their chromosomal localization, sequence features, and gene structure, laying the foundation for studying the functions associated with this gene family.

Salt stress is one of the major abiotic stresses affecting potato growth in agricultural production. Under adverse stress conditions, significant changes occur in the internal physiology and external morphology of the potato, leading to reduced tuber quality and yield [23]. Researchers have made significant discoveries regarding the involvement of pectate lyase gene family members in plant growth and development, fruit softening, and disease resistance. However, there are no reports on the response of members of the pectate lyase gene family to salt. Therefore, this study selected a diploid potato material and conducted salt stress treatment under tissue culture conditions. The results showed that 27 family members had different responses.

In summary, this research is based on the DM v6.1 Genome assembly for the doubled monoploid potato information. It identifies and analyzes the members of the *StPL* gene family, while also examining the expression of these genes under salt stress. These findings provide a basis for the further analysis of the differentiation process and biological function of the *StPL* gene family in potato. These results laid the foundation for further analyzing the differentiation process and biological functions of the *StPL* gene family in potato and provided a theoretical basis for the discovery and screening of stress resistance genes in potato.

## 2. Results

### 2.1. Chromosomal Localization of StPL Genes

A total of 93 candidate genes with “pectate lyase” annotations were obtained through the term’s search in SpudDB database (http://spuddb.uga.edu/, accessed on 10 November 2023), and 27 *PL* family members were ultimately identified utilizing NCBI online software (https://www.ncbi.nlm.nih.gov/Structure/cdd/cdd.shtml, accessed on 10 November 2023) to exclude sequences lacking the conserve domains of Pec_lyase_C and Pec_lyase_N. Based on their locations on the chromosomes, they were named *StPL1*-*27*. As shown in Figure 1, the 27 *StPL* genes were unevenly distributed on chromosomes 1 to 6, 9, 11, and 12. Among them, chromosome 2 had the largest number of *PL* genes (eight genes), followed by chromosome 3 (five genes). There was only one gene each on chromosomes 11 and 12 (*StPL26* and *StPL27*). *StPL* genes on chromosomes 4, 5, 6, 9, 11, and 12 were predominantly located at the distal ends of the chromosomes.

### 2.2. Physicochemical Properties and Subcellular Localization of the StPL Proteins

Physicochemical property analysis and subcellular localization prediction of StPL members were performed using the online tools ProtParam and WoLF PSORT (Table 1). The results revealed that the length of the family in potato varied from 83 (StPL2) to 1332 amino acids (StPL16). Ten out of the twenty-seven members encode acidic proteins, while the remaining members encode neutral or basic proteins. Additionally, the molecular mass of the StPLs ranged from 150.18 kDa (StPL16) to 9.53 kDa (StPL2). Except for StPL1 and StPL2, whose instability index was greater than 40, the rest of the proteins had values below 40, suggesting that members were relatively stable. The analysis based on the predicted values of the grand average of hydropathicity showed that all the values were negative, and the aliphatic index varied from 68.52 to 92.89, indicating that most of these proteins exhibit some level of hydrophilicity. Eight StPLs were localized to the extracellular matrix, five to the vacuole, four to the chloroplast, and three to the cytoskeleton. Three members were localized to the plasma membrane, two members were localized to the peroxisome, and only StPL16 was localized to the mitochondria.

The above results indicate that the molecular mass of different members varied and were widely distributed. In addition, the hydrophilicity of all StPLs was less than 0. Therefore, it is hypothesized that the proteins in this family may be hydrophilic.

### 2.3. Gene Duplication and Collinearity Analysis of StPL Gene Family Members 

The *StPL* family genes were analyzed for covariance using MCScanX sequencing in TBtools (Figure 2). The results revealed that 15 genes were tandemly duplicated, and there were also segmentally duplicated genes with covariance. Most of the covariate genes were located on chromosomes 2, 3, 5, and 6, with the highest number found on chromosome 2, totaling six pairs. On chromosome 1, *StPL1* was identified as an independent tandem duplication gene, while the remaining genes were not only tandem duplicated but also fragment duplicated. This indicates that both tandem duplications and fragment duplications play a role in the expansion of the *StPL* gene family.

### 2.4. Gene Duplication and Collinearity Analysis of PL Gene Family Members in Potato and Arabidopsis

Co-linearity mapping of the *AtPLL* gene of *A. thaliana* and the *StPL* gene of potato revealed that chromosome 2 had the greatest number of co-linear genes, with 12 pairs. *StPL3* was co-linear with *AtPLL8*, 10, and 11; *StPL5* was co-linear with *AtPLL4* and 6; *StPL7* was co-linear with *AtPLL8*, 9, 10, and 11; and *StPL9* was co-linear with *AtPLL10*, 15, and 16. It indicates that there is some similarity between the *PL* genes of potato and *A. thaliana* (Figure 3).

### 2.5. Phylogenetic Tree Analysis of StPL Gene Family Members 

Referring to the previous classification of *PL* genes in *A. thaliana* [16], rice [15], and tomato [18], we divided the 27 *PL* genes into eight groups, with Group 1 having the most members at 11. *AtPLL12*, *AtPLL13*, *AtPLL23*, and *AtPLL26* in Group 1, as well as *AtPLL1* in Group 5, induce cell elongation and differentiation through auxin [15,24]. *SlPL*, which belongs to the same branch as *StPL13* and *StPL20*, can be induced by auxin, abscisic acid, and ethylene. Inhibition of its expression will increase the pectin content of tomato fruits, delay fruit rot, and extend the storage period. Therefore, it is hypothesized that *StPL13* and *StPL20* may be associated with potato fruit hardness [22]. *AtPLL13* (*PMR6*) is in the same group as *StPL18*. The *PMR6* mutation causes plants to dwarf, leaves to become small and rounded, and results in increased resistance to powdery mildew in *pmr6* plants [21]. The knockout of *StPL18* causes morphological changes in potato plants [25]. Many *PL* genes are related to flower development. Silencing *OsPLL4* results in abnormal pollen microspores, while *StPL21*, *StPL24*, and *StPL25* belong to the same evolutionary branch as *OsPLL4.1* and *OsPLL4.2*. Mutations in *StPL21*, *StPL24*, and *StPL25* are speculated to affect the floral organ morphology of potato. Additionally, silencing *OsPLL3* in Group 6 leads to pollen degradation, which affects the development of floral organs and spikelets [15]. *StPL5* and *StPL6* were assigned to Group 3, which consisted of seven members. *AtPLL4* and *AtPLL7*, also part of this group, showed high expression in the floral organs. It was speculated that *StPL5* and *StPL6* were associated with flower development [19], while the remaining members in the evolutionary tree had not been functionally characterized. The above results suggest that *StPL* genes are more closely related to *PL* genes in other species (Figure 4).

### 2.6. StPL Gene Family Members’ Gene Structure and Conserved Motif Analysis

The gene structure and conserved motif analysis of *StPL* members were conducted using TBtools, revealing differences in the number and types of motifs present in *StPL*. Six differently positioned and relatively conserved motifs were screened. Most members contain six motifs. *StPL2* does not contain any motifs. *StPL7* has only one motif. *StPL15* has two motifs. *StPL23* has three motifs, and *StPL11* has five motifs. Taken together, these motifs and their arrangements show a high degree of conservation in the *StPL* family (Figure 5a). Screening of the NCBI-CDD [26] conserved domains revealed that all *StPLs*, except *StPL2*, have the complete Pec_lyase_C domain. *StPL2* has the Pec_lyase_N domain, further identifying all members as belonging to the pectate lyase family (Figure 5b). *StPL1*, *StPL3*, *StPL4*, *StPL5*, *StPL10*, *StPL11*, *StPL14*, *StPL19*, and *StPL27* have no UTRs, which may have implications for mRNA transport, stability, translational regulation, and incomplete gene annotation [27].

### 2.7. Analysis of the Cis-Acting Elements in the Promoter of StPL Gene Family Members

The cis-acting element analysis of *StPL* promoter regions by the PlantCARE database revealed the presence of numerous hormone response elements in 27 *StPL* promoter regions (Figure 6). These elements include the auxin response element, abscisic acid response element, methyl jasmonate (MeJA) response element, and gibberellin response element. MeJA is a significant phytohormone abundant in plants, playing a crucial role in various physiological processes like flowering, fruiting, and senescence [28]. It has been found that the exogenous application of MeJA elicits a defense response in plants, suggesting that *StPL* may be involved in the plant’s response to adversity stress through MeJA and wound response elements [29]. In addition, defense and stress response elements, low-temperature response elements, and MYB binding sites involved in drought inducibility response elements were present in the promoter regions of the *StPL* genes. Many light-responsive elements were also found on the promoters of all the *StPL* genes, further suggesting that the *StPL* genes are involved in the plant’s stress response.

### 2.8. Expression Pattern Analysis of StPL Gene Family Members under Salt Stress

To investigate the response of *StPL* genes when plants encountered salt stress, we treated potato seedlings under single (Figure 7a) and mixed salt (Figure 7b) conditions. Then, we analyzed the expression of each gene in the transcriptome data and utilized TBtools to generate a differential expression heatmap. The results indicated that most of the members were up-regulated under salt stress, suggesting that members of the family play a role in the potato’s response to salt. Under single salt stress, *StPL3*, *StPL4*, *StPL5*, and *StPL10* were not expressed. Except for *StPL1*, *StPL7*, and *StPL23*, whose expression decreased under single salt stress, the expression of the other members increased to varying degrees. *StPL10* and *StPL11* were not expressed under mixed salt stress, while the expression of *StPL2*, *StPL12*, *StPL13*, *StPL17*, *StPL20*, *StPL21*, *StPL22*, *StPL23*, *StPL24*, *StPL25,* and *StPL26* was up-regulated with the duration of treatment under mixed salt stress. This suggests that although the *StPL* family members play a role in salt stress, there may be a division of labor among them.

In addition, we observed that the expression of *StPL18* increased with treatment under both single and mixed salt stress conditions. Its expression was notably distinct from that of other members. Yang discovered that the knockout of the *StPL18* gene led to potato plants with more numerous dwarfed branches and yellowing leaf margins, ultimately impacting the yield [25]. Therefore, the *StPL18* gene will be selected for further study.

### 2.9. Construction of Overexpression Vector of Potato StPL18 Gene and Subcellular Localization in Tobacco

To gain a deeper understanding of the *StPL18* gene function, both upstream and downstream primers were designed. The reverse primer was designed to exclude the termination codon. PCR amplification was performed using the constructed *pEASYT1-StPL18* as a template to produce electrophoretic bands of the expected size. The resulting products were purified, recovered, and subsequently used for ligation and transformation through homologous recombination with the purified and recovered super1300, utilizing *Xba*I and *Kpn*I double digestion. Linear vectors were ligated for transformation, and positive colonies were screened, identified by PCR, and sent for sequencing after proper digestion and identification (Appendix A). Under confocal microscopy, the target gene was found to overlap with the endoplasmic reticulum marker in tobacco leaves, indicating that StPL18 is localized in the endoplasmic reticulum (Figure 8).

## 3. Discussion

### 3.1. Analysis of Pectate Lyase Gene Family Members

Potato is the most important tuber crop. Members of the pectate lyase gene family play crucial roles in the growth and development of potato. In this study, 27 members were identified in potato. The physicochemical properties of the StPL indicate that the proteins encoded by most members are hydrophilic. Fragment duplication and tandem duplication are present within the *StPL* gene family. A total of 15 genes are tandem duplicated and fragment duplicated genes. A comparison with *A. thaliana* genes reveals that not only fragment duplication but also rearrangement of some chromosomes at different positions may occur among different chromosomes. Six motifs were present in most *StPLs*, suggesting that *StPLs* may exhibit relatively conserved functions during potato growth. Evolutionary tree analysis showed that *StPL* is closely related to *PL* from other species; this indicates that the *PL* gene is functionally conservative in different species. In addition, gene structure analysis revealed a high diversity of exon numbers in *StPLs*, which may be related to their functional diversity.

### 3.2. Expression Pattern of Pectate Lyase Gene Family Members under Salt Stress

Pectin in plant cell walls carries a negative charge and can bind to cations such as Ca^2+^ and Na^+^, enhancing the hardness of the cell wall [30]. It has been shown that in high-salt environments, excess Na^+^ may disrupt the normal cross-linking of pectin and inhibit cell elongation by replacing Ca^2+^ bound to pectin [31]. Pectin synthesis and degradation in plants are correlated with salt tolerance. *Sos6* mutants, defective in the *A. thaliana CSLD5* gene involved in pectin biosynthesis, impede xylan and pectin synthesis in the cell wall. This leads to the increased susceptibility of the mutants to drought and salt stress [32]. Meanwhile, the degree of pectin methyl esterification is closely related to the hardness of plant cell walls. Plants regulate the level of pectin methyl esterification primarily through pectin methyl ester transferases (PMEs) and pectin methyl esterase inhibitors (PMEIs) [33]. Meanwhile, salt stress can activate PMEs and promote the demethylation of pectin in the cell wall, thereby initiating downstream salt stress mechanisms [34]. It has been shown that the *SlPL-RNAi* fruits of tomato have increased firmness, higher hemicellulose content, and greater resistance to rot and disease [18]. In this study, we analyzed the promoter region of *StPL* genes and found that most of the genes were up-regulated in single and mixed salt stress, indicating that these genes may be involved in the response of potato to salt stress. Yang demonstrated that *stpl18* is a yellow margin (YM) gene through knockout. Its mutation resulted in the yellowing of the edges of potato leaves, a decrease in single potato weight, an increase in branching, and an increase in the number of potato tubers [25]. From these results, the expression of *StPL18* increased with the prolongation of salt stress. During previous research, the mutation of *PMR6*, the homologous gene of that in *A. thaliana*, resulted in smaller plants, increased pectin content, enhanced resistance to powdery mildew, and affected stomatal movement [21]. It was found that *PMR6* influenced the size of *A. thaliana*’s rosette by impacting cell expansion and proliferation. The homologous gene of *StPL18* in rice, *DEL1*, serves the same function in cell cycle progression and leaf growth. The *DEL1* mutation led to a reduction in the number of cell cycles, ultimately resulting in dwarf plants due to the blockage of the G1 to S phases of the cell cycle [35]. It is hypothesized that the *stpl18* mutant plants may have increased salt tolerance due to decreased gene expression causing increased pectin content.

The subcellular localization of StPL18 indicates that it may be located in the endoplasmic reticulum. The endoplasmic reticulum is where all secreted proteins and most membrane proteins are synthesized and folded. When plants face adversity, misfolded and unfolded proteins accumulate in large quantities in the endoplasmic reticulum, causing endoplasmic reticulum stress [36]. Elucidating the endoplasmic reticulum stress response signaling pathway in plants provides a basis for understanding the mechanisms involved in responding to plant adversity. Endoplasmic reticulum stress is closely related to various environmental stresses. The up-regulated expression of the *StPL18* gene may be subject to the endoplasmic reticulum stress response.

## 4. Materials and Methods 

### 4.1. Identification of Members of the Potato PL Gene Family

A search for the keyword “Pectate lyase” in the potato genomics resources SpudDB database (http://spuddb.uga.edu, accessed on 10 November 2023), which identified 93 genes annotated with pectate lyase, and their gene IDs were obtained. Then, download the genome assembly for the doubled monoploid potato DM 1-3 516 R44—v6.1 and genome annotation: DM_1-3_516_R44_potato. v6.1. hc_gene_models. gff3 files from the database. Then, utilize TBtools software V2.086 [37]. Within the software, navigate to the Sequence Toolkit, click on GFF3/GTF Manipulate, select GXF Sequence Extract, and proceed to insert the GFF3 file. Finally, extract all gene CDS sequences of potato. Open Fasta tools and then Fasta Extract (recommended). Input the gene ID and CDS file to extract the CDS of *StPL*. The extracted CDS is translated into protein using the Sequence Toolkit, ORF Prediction, and Batch Translate CDS to Protein. Upload the protein sequence to the NCBI Batch Web CD-Search Tool [38], screen for the structural domains, and only keep proteins containing the Pec_lyase_C and Pec_lyase_N structural domains with Pfam accession numbers PF00544 and PF04431. After screening, a total of 27 *StPL* genes were obtained.

### 4.2. Chromosomal Localization of Potato StPL Genes

In TBtools, open the “Graphics” tab, select “Show Genes on Chromosome”, choose “Gene Location Visualize from GTF/GFF”, import the GFF3 file of the potato and the ID of the *StPL*, enter the “Rename ID” and the “Chromosome Sort ID”, and then click “Start”.

### 4.3. Analysis of Physicochemical Properties of Potato StPL Protein and Prediction of Subcellular Localization 

Protein sequences in FASTA format were uploaded using the online websites ProtParam (https://web.expasy.org/protparam, accessed on 10 November 2023) [39] and WoLF PSORT (https://wolfpsort.hgc.jp/, accessed on 10 November 2023) [40] to obtain amino acid length, molecular mass, theoretical isoelectric point, instability index, lipid solute number, hydrophilicity results, and predictions of subcellular localization.

### 4.4. Phylogenetic Tree and Gene Structure Analysis of the Potato StPL Gene

The *PL* gene sequences of tomato, *A. thaliana*, and rice were obtained from the articles by Lu Yang [18], Lingxia Sun [16], and Yinzhen Zheng [15]. The MEGAX software V10.2.6 was used to construct the evolutionary tree. Firstly, the FASTA file was imported, and the meg format file was saved after sequence comparison. The meg file was then imported, and the “Construct/Test Neighbor-Joining Tree” option under PHYLOGENY was selected with the following parameter settings: the Bootstrap value was set to 500, and the model for genetic distance calculation was set to P-distance. Partition deletion was set to 50%, while the rest of the parameters were kept as default. The evolutionary tree was further enhanced using ITOL v6 (https://itol.embl.de, accessed on 10 November 2023), online software [41].

Use TBtools, click on “Others”, then select “MEME Suite Wrapper”, and finally choose “Simple MEME Wrapper”. Upload protein sequences from StPLs and set the output path. The number of motifs to find is set to 6, and the maximum E-value is set to 0.0001. The mode selection is set to zero or one occurrence per sequence to obtain meme xml file.

Use NCBI-CDD to obtain the prediction result of the conservative domain. Download Excel sheet file.

Using TBtools, select Gene Structure View, insert the protein ID, potato GFF3 file, motif xml file and NCBI-CDD excel sheet file, then click Start to analyze, and obtain the results.

### 4.5. Gene Collinearity Analysis

Open TBtools, select “Others”, then choose “Plugin”, followed by “Plugin Store”. Check “One Step MCScanX [Super-Fast] by CJ” in the menu bar and click the “Install” button to install the selected plugin. Select “Others” > “Plugin” > “Plugin Store”, then choose “One Step MCScanX [Super-Fast] by CJ” from the menu bar and click on the “Install Selected Plugin” button to install it. The genome files of *A. thaliana* and potato were imported using the One Step MCScanX plugin, and the gene covariance schematic was generated.

### 4.6. Analysis of Potato StPL Promoter Cis-Acting Elements

Open TBtools, go to Graphics, select Gene Structure View (Advanced), import *StPL* gene ID, DM V6.1 GFF3 file, ID rename file, and click on start. Use TBtools and the GTF/GFF3 Sequence Extract tool to extract the 2000 bp upstream of all coding sequences (CDS) of potato. Utilize Quick Fasta Extractor or Filter to extract the 2000 bp upstream of the CDS of the *StPL* gene. Then, open Fasta Sequence Manipulator to convert the sequence to uppercase. Convert the sequence to uppercase. The extracted files were submitted to the PlantCare website (http://bioinformatics.psb.ugent.be/webtools/plantcare, accessed on 10 November 2023) [42]. For predicting the cis-acting elements. The predicted table files were analyzed using Excel, then imported into TBtools. Subsequently, the Graphics option was selected, followed by the Simple BioSequence Viewer to draw the schematic diagram of promoter cis-acting elements.

### 4.7. Analysis of the Expression Pattern of StPL Family Genes in Potato Seedlings under Single and Mixed Salts

The plant material used in this study is diploid potato 320-02. This material is closely related to the reported diploid potato RH, both belonging to the *Tuberrosum* subspecies of potato. A total of 320-02 were subjected to different time-lengths of NaCl (100 mmol/L) single salt and equal volume mixed-salt treatments of NaCl + Na_2_SO_4_ (100 mmol/L), respectively; the processing time points were 0, 6, 24, and 48 h, with three replicates per time point. 

Take samples for transcriptome sequencing.

The resulting transcriptome data were compared to the potato reference genome DM (V6.1) using HISAT2 (V2.2.1) to construct an index. HISAT2 compared the reads of each sample to the genome to obtain SAM files, which were then sorted and converted to BAM files using Samtools [43]. Splicing transcripts used StringTie software (V2.2.3) and conducted a reference transcriptome analysis to estimate transcript expression and use R language script Ballgown for further analysis [44,45]. The gene expression levels were calculated according to the FPKM method. The heatmap of gene expression of pectate lyase gene family was produced using TBtools.

### 4.8. Subcellular Localization of StPL Genes

To obtain the subcellular localization information of StPL proteins, a fusion protein expression vector of *StPL18* and GFP was constructed. GFP was utilized as a reporter gene, and the position of GFP was observed to determine the location of *StPL* proteins. The recombinant protein, with GFP fused to the coding region of the *StPL* gene and the stop codon removed, was expressed using the 35S promoter. The subcellular localization of the protein was analyzed later. Sper was employed as an endoplasmic reticulum localization signaling protein to construct a fusion protein with the red fluorescent protein mKATE [46]. 

Experimental Steps

(1)Pick Agrobacterium containing the corresponding vectors and single colonies of *Agrobacterium* containing the P19 plasmid. Inoculate them into 10 mL of LB liquid medium with the corresponding resistance and incubate them for 24 h at 28 °C and 220 rpm until the OD600 value greater than 0.6.(2)Centrifuge at 5000 rpm for 2 min and discard the liquid medium.(3)Resuspend the organisms by adding 700 μL of tobacco infestation buffer, mix well, and take 200 μL of the bacterial solution to measure the OD600 value.(4)Mix *Agrobacterium* with different vectors in a 2 mL centrifuge tube so that the OD600 value of Agrobacterium containing the P19 plasmid is adjusted to 0.3, and other *Agrobacterium* is adjusted to 0.5. Leave the mixture at room temperature for about 3 h. Afterward, carefully pipette the well-grown *Agrobacterium* from the center of the centrifuge tube and discard the rest.(5)Take the tobacco leaves (tobacco has about four true leaves and is 10 cm tall) with good growth conditions and inject *Agrobacterium* from the back of the leaves using a 1 mL syringe until the leaves are completely wet.(6)Incubate in the dark for 24 h and then transfer to dim light culture for 36 h.(7)For co-localization, the marker plasmid was transformed into *Agrobacterium*, operating in suspension with the constructed vector plasmid *Agrobacterium*, and mixed at an equivalent ratio before injection into tobacco leaves.

## 5. Conclusions

By analyzing the *PL* gene family in potato, *StPL* genes were classified into eight subfamilies based on phylogeny and chromosomal localization. All *StPLs* had a conserved Pec_lyase_C domain and diverse gene structures. Expression analysis revealed that *StPLs* exhibited different expression patterns under various salt treatments. These results suggest that *StPLs* play important roles in plant growth, development, and stress response. 

In this study, 27 *StPL* genes in potato were screened using salt treatments. Eventually, the StPL18 protein, located in the endoplasmic reticulum, garnered attention. It was discovered to regulate the morphogenesis of potato, as previously researched, and *StPL18* was up-regulated under different salt treatments. The elevated expression of pectin cleavage enzymes destabilizes the cell wall structure, compromising its protective function. This allows harmful ions like Na^+^ to penetrate the cell, increasing the plant’s salt sensitivity. This sensitivity can be heightened by acquiring material from the *pl* mutant. Mutations in the *PL* gene impact the weight of potato tubers but result in more branches and tubers, which is advantageous for certain consumers preferring smaller tubers. Therefore, this gene may hold significant value in enhancing potato salt tolerance and cultivating new varieties.

## Figures and Tables

**Figure 1 plants-13-01322-f001:**
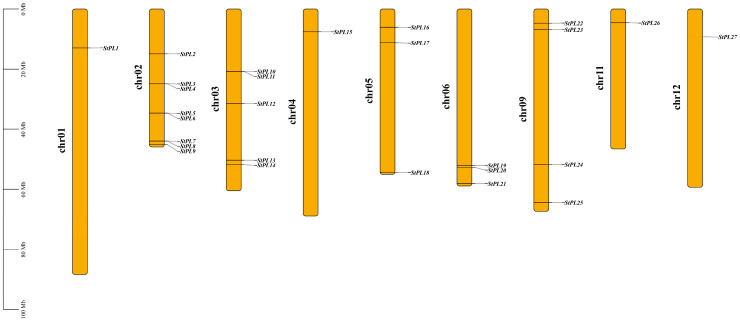
Genomic distribution of *StPLs* on potato chromosomes. Chromosome numbers are on the left and *StPLs* are on the right of chromosomes. Scale bar on the left indicates chromosome length.

**Figure 2 plants-13-01322-f002:**
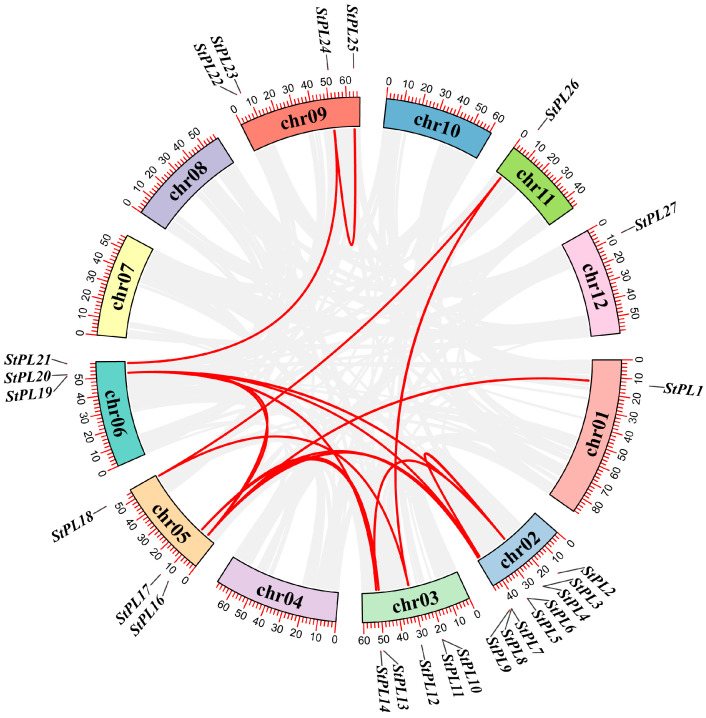
The collinearity analysis of *StPL* genes is represented via a Circos map, exhibiting the relationships among the gene pairs. Gene pairs are represented by red lines.

**Figure 3 plants-13-01322-f003:**
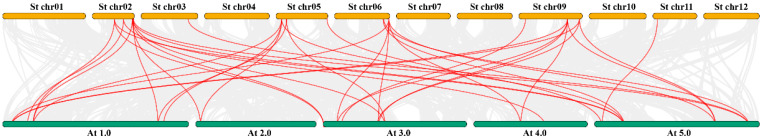
Collinearity analysis of the *PL* gene family in potato and the *PL* gene family in *Arabidopsis*. The red lines represent genes within these two families that have collinearity relationships. The gray color represents genes with no covariate relationship.

**Figure 4 plants-13-01322-f004:**
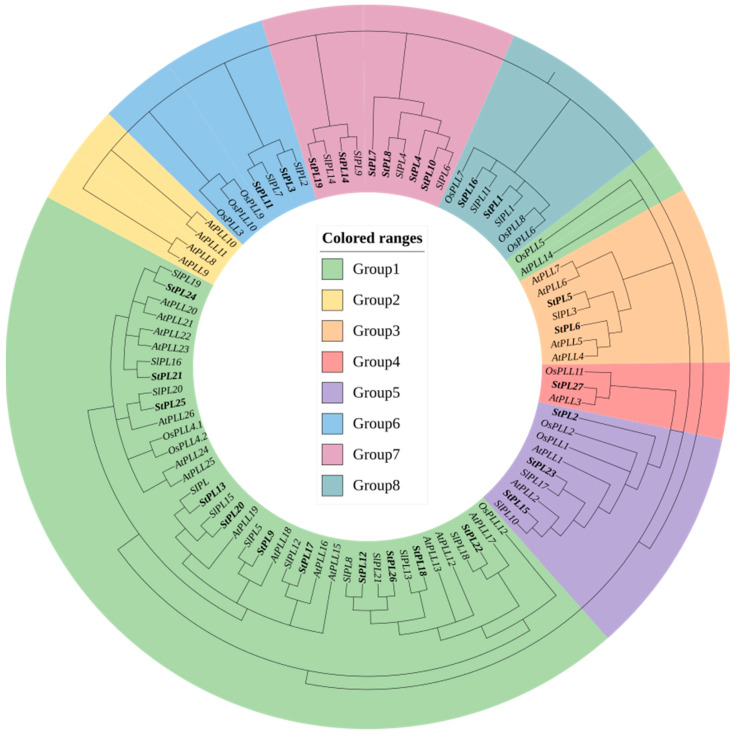
Neighbor joining phylogenetic tree of potato (St), *Arabidopsis* (At), tomato (Sl), and rice (Os), where *StPL* in potato is highlighted. The different colors on the outside of the tree represent the corresponding groups, where green, yellow, orange, red, purple, blue, light purple, and cyan represent Group 1, Group 2, Group 3, Group 4, Group 5, Group 6, Group 7, and Group 8, respectively.

**Figure 5 plants-13-01322-f005:**
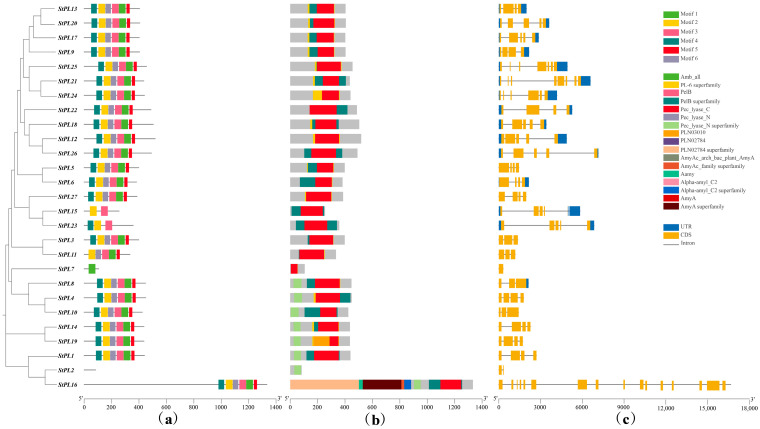
Gene structure diagram of *StPL* gene family. (**a**) Conserved motifs in the family, where different colors represent different motifs. A maximum of 6 motifs were searched with motif length between 10 and 50 residues. Motif occurrence was set to zero or one occurrence per sequence. (**b**) Conserved domains possessed by members of the family, where different colors represent different conserved domains; (**c**) Gene structure of *StPLs*, where blue square represents UTR and yellow square represents CDS, the line represents Intron.

**Figure 6 plants-13-01322-f006:**
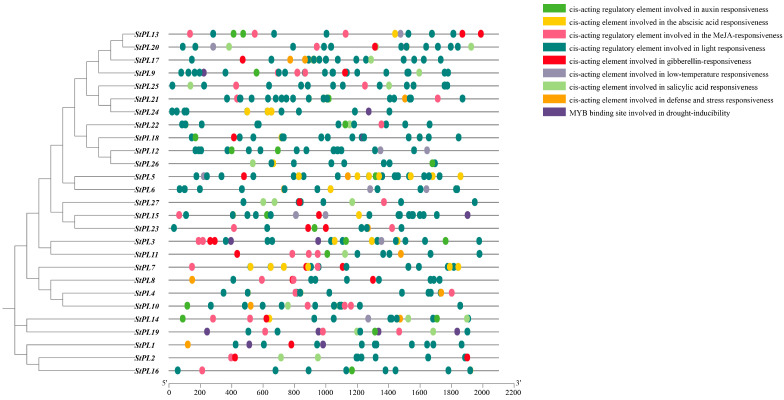
Cis-acting elements predicted in 2000 bp sequence upstream of the *StPL* gene. Different colored squares represent various types of cis-acting elements.

**Figure 7 plants-13-01322-f007:**
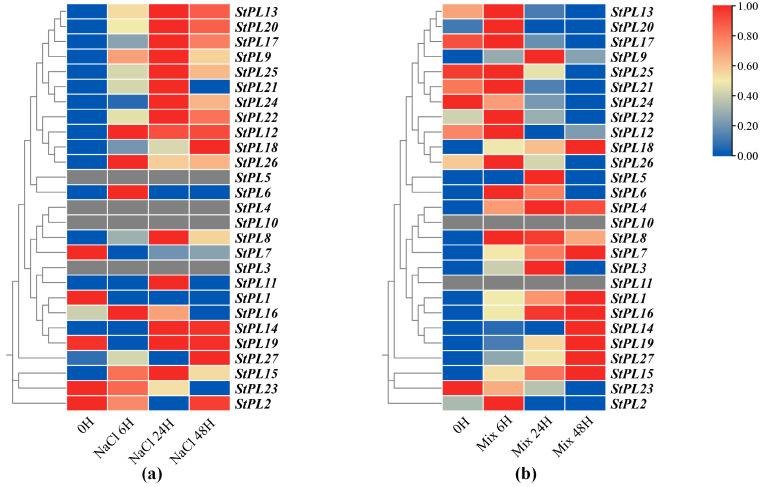
Expression patterns of the members of the potato *StPL* gene family under single salt and mixed salt stress. (**a**) Expression levels of the members of the family under single salt; (**b**) expression levels of the members of the *StPL* family under mixed salt. The color bar in the figure represents the log2 expression values, and the expression values were normalized using TPM + 1.

**Figure 8 plants-13-01322-f008:**
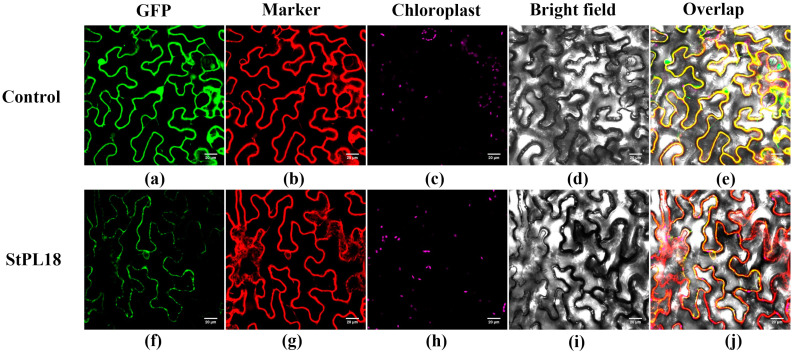
Subcellular localization of StPL18. Transient expression of control (35S::GFP) and 35S::StPL18:GFP fusion protein in tobacco leaves. GFP alone (negative control) constructs were transiently expressed under the cauliflower mosaic virus 35S promoter, and the GFP signal was observed by confocal microscopy 72 h after transfection. (**a**) Super-1300-GFP vector control; (**f**) localization of Super-1300-StPL18-GFP fusion protein; (**b**,**g**) endoplasmic reticulum membrane was marked by Sper; (**c**,**h**) chlorophyll autofluorescence; (**d**,**i**) bright field; (**e**,**j**) the overlap images are shown; the scale bars are 20 μm.

**Table 1 plants-13-01322-t001:** Physiological properties and subcellular localization of the *StPL* gene family.

Gene ID	Transcript ID	Molecular Mass (Da)	Amino Acid Length (aa)	Theoretical pI	Instability Index	Stability	Aliphatic Index	Grand Average of Hydropathicity	Subcellular Localization
*StPL1*	Soltu.DM.01G009330.1	49,343.26	439	8.24	40.13	unstable	81.55	−0.349	Chloroplast
*StPL2*	Soltu.DM.02G003590.1	9534.89	83	5.27	48.63	unstable	92.89	−0.125	Chloroplast
*StPL3*	Soltu.DM.02G009950.1	44,342.64	397	8.67	36.01	stable	81.08	−0.344	Peroxisome
*StPL4*	Soltu.DM.02G009960.1	50,669.45	448	7.65	37.59	stable	70.09	−0.45	Extracellular
*StPL5*	Soltu.DM.02G020450.1	44,279.08	397	6.86	33.21	stable	77.36	−0.321	Extracellular
*StPL6*	Soltu.DM.02G020460.1	42,099.02	381	9.11	29.28	stable	79.37	−0.267	Extracellular
*StPL7*	Soltu.DM.02G031990.1	11,791.5	105	8.58	35.78	stable	75.14	−0.098	Extracellular
*StPL8*	Soltu.DM.02G032040.1	49,905.67	446	8.31	28.17	stable	76.3	−0.345	Vacuole
*StPL9*	Soltu.DM.02G033640.1	44,508.98	403	7.65	38.52	stable	77.67	−0.382	Extracellular
*StPL10*	Soltu.DM.03G008540.1	48,094.64	423	8.59	34.61	stable	70.76	−0.522	Cytoplasmic
*StPL11*	Soltu.DM.03G008580.1	36,799.61	333	6.22	31.4	stable	86.4	−0.266	Cytoplasmic
*StPL12*	Soltu.DM.03G011300.1	56,927.17	517	6.38	31.8	stable	83.35	−0.269	Plasma membrane
*StPL13*	Soltu.DM.03G025090.1	44,178.86	403	6.96	35.52	stable	78.88	−0.242	Extracellular
*StPL14*	Soltu.DM.03G027030.1	48,610.72	435	9.51	34.68	stable	79.1	−0.32	Vacuole
*StPL15*	Soltu.DM.04G007160.1	28,596.42	254	8.67	27.68	stable	80.24	−0.416	Cytoplasmic
*StPL16*	Soltu.DM.05G006330.1	150,181.53	1332	6.05	43.29	unstable	78.68	−0.475	Mitochondrial
*StPL17*	Soltu.DM.05G010370.1	44,248.03	401	6.93	33.57	stable	77.83	−0.3	Extracellular
*StPL18*	Soltu.DM.05G026200.1	55,513.46	503	5.9	39.15	stable	83.7	−0.162	Membrane
*StPL19*	Soltu.DM.06G026130.1	48,636.13	435	8.78	35.83	stable	76.67	−0.391	Vacuole
*StPL20*	Soltu.DM.06G026970.1	44,496.3	405	7.64	31.07	stable	84.25	−0.218	Vacuole
*StPL21*	Soltu.DM.06G033950.1	48,183.37	434	6.8	45.92	unstable	80.94	−0.231	Chloroplast
*StPL22*	Soltu.DM.09G005200.1	53,964.23	486	5.48	37.44	stable	75.99	−0.324	Vacuole
*StPL23*	Soltu.DM.09G007270.1	40,151.06	357	7.92	36.37	stable	68.52	−0.602	Peroxisome
*StPL24*	Soltu.DM.09G018370.1	48,797.13	439	7.68	36.78	stable	79.54	−0.274	Chloroplast
*StPL25*	Soltu.DM.09G028450.1	50,015.35	454	7.28	34.07	stable	74.56	−0.299	Extracellular
*StPL26*	Soltu.DM.11G004550.1	53,705.64	490	5.95	29.56	stable	84.37	−0.125	Plasma membrane
*StPL27*	Soltu.DM.12G009730.1	42,657.73	385	9.52	28.18	stable	73.61	−0.382	Vacuole

## Data Availability

Data are contained within the article.

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
