# Peer review of "Genome-Wide Identification of the Pectate Lyase Gene Family in Potato and Expression Analysis under Salt Stress"

_plants, 2024, doi:10.3390/plants13101322_

Round 1

Reviewer 1 Report

Comments and Suggestions for Authors

 The manuscript “Genome-Wide Identification of the Pectin Lyase Gene Family in Potatoes and Expression Analysis under Salt Stress” by Wang et al. characterizes the pectin lyase gene family in potatoes. It is mostly a bioinformatics study, but also has a gene localization experiment, the conclusions of which however do not convince me. 

The computational analysis includes transcriptomics data analysis, but it is somewhat misleadingly presented as if it was done in this study, while it appears to be done previously, in which case it should be properly cited. 

Some of the bioinformatics analysis is contradictory, e.g. if the pectin lyases are hydrophilic or hydrophobic. I believe the paper requires careful revisions (see points below).

These two sentences are repeated:

“During the growth and development of the potato, the organ grows from small to large, and the plant grows from low to high. The intrinsic structural changes mainly involve an increase in the size of the cells and an increase in the number of cells[4].

During the growth and development of the potato, the tuber enlarges from small to large, while the plant grows from low to high. The intrinsic structural changes mainly involve an increase in the size of the cells and an increase in the number of cells[5].”

“They do not function independently but are interlinked to form a highly complex and dynamic  network, which collectively determine the mechanical properties of the cell wall[6].”

unclear what “they” refers to, rephrase for clarity .

“Through conservative domain analysis, all plant PL proteins contain” Plant proteins don’t contain anything “through analysis.” Re-pharase e.g. by adding “it was revealed that” after analysis.

“Members of the Pectin Lyase gene family have made important discoveries …”; it is not the pectin lyases that made important discoveries.

“A total of 93 candidate genes with pectin lyase annotations were obtained through a search” be more specific; briefly mention what kind of search, even though you state more details in the Methods section.

“Among the 27 StPL gene family members, 110 were acidic amino acids,” I don’t understand what this means. How can gene family members be amino acids?

“while the remaining were neutral or basic amino acids” Same question as above.

“and all StPL  family members consisted of hydrophilic amino acids” consisted is a strong word; there were surely also some non-polar amino acids? How about “consisted mainly of”? Or “contained mostly”?

“proteins encoded by genes in this family may be hydrophobic.” Hydrophobic proteins that consist of hydrophilic amino acids? Clarify

“Figure 6. Cis-acting elements of the potato StPL family, different colors represent different cis-acting elements.” Well, these are “putative” cis-acting elements; they may or may not be significant. If possible, include any confidence estimate, such as E or P values.

“We analyzed the expression of each gene in the transcriptome data” mention here how you generated the transcriptome data; cite previous study if appropriate.

“Figure 7.” What is the unit of the legend? What is  “0 to 1”  Is it supposed to be log2 fold change, or anything else? State this clearly in the figure. 

Figure 8. this could be moved to supplemental material.

Figure 9. “Under confocal microscopy, the cell signal was localized to the membrane system, leading to speculation that StPL18 might be situated in the endoplasmic reticulum (Figure 9e).” It does not look like ER to me; not sure what it is, Golgi? Compare to published dGFP images, or use some known markers yourself for comparison. What does the signal peptide indicate? Using e.g. DeepLoc2.0 could help sorting this out.

Also, describe the experiment in more detail, what is the tissue and plant we are looking at?

“consist of hydrophilic amino acids, which are chemically unstable.” The amino acids are unstable? I don’t understand this.

“It was speculated that StPL18 might act as an activator of specific defense responses, enhancing the sensitivity of potatoes to salt stress.” Where was this speculated, add citation. Also, why would activation of defense enhance the sensitivity to salt stress? Explain.

“The subcellular localization of StPL18 indicates that it may be situated in the endoplasmic reticulum.” Again, I’m not convinced of this.

Materials and Methods

“ Search for the keywords "Pectate lyase"” but the paper was on “pectin lyases. Which is it?

“Potato materials were subjected to different time-length NaCl (100 mmol/L) single salt and equal volume mixed salt treatments of NaCl+Na2SO4 (100 mmol/L), respectively, to analyze the expression of each gene member of the family in the transcriptome data”. I don’t understand how the transcriptome data was generated. After the plants were subjected to salt, what was done next? Or is this data-mining? In that case, don’t present it like plants were actually treated in this study.

Comments on the Quality of English Language

Moderate revisions required.

Author Response

The manuscript “Genome-Wide Identification of the Pectin Lyase Gene Family in Potatoes and Expression Analysis under Salt Stress” by Wang et al. characterizes the pectin lyase gene family in potatoes. It is mostly a bioinformatics study, but also has a gene localization experiment, the conclusions of which however do not convince me.

The computational analysis includes transcriptomics data analysis, but it is somewhat misleadingly presented as if it was done in this study, while it appears to be done previously, in which case it should be properly cited.

Some of the bioinformatics analysis is contradictory, e.g. if the pectin lyases are hydrophilic or hydrophobic. I believe the paper requires careful revisions (see points below).

These two sentences are repeated:

“During the growth and development of the potato, the organ grows from small to large, and the plant grows from low to high. The intrinsic structural changes mainly involve an increase in the size of the cells and an increase in the number of cells[4].

During the growth and development of the potato, the tuber enlarges from small to large, while the plant grows from low to high. The intrinsic structural changes mainly involve an increase in the size of the cells and an increase in the number of cells[5].”

Reply:We are so sorry for this mistake, we have removed the repeated sentence.

“They do not function independently but are interlinked to form a highly complex and dynamic network, which collectively determine the mechanical properties of the cell wall[6].”

unclear what “they” refers to, rephrase for clarity.

Reply:“They do not function independently but are interlinked to form a highly complex and dynamic network, which collectively determine the mechanical properties of the cell wall” “they” refers to “pectin, hemicellulose and cellulose”.

“Through conservative domain analysis, all plant PL proteins contain” Plant proteins don’t contain anything “through analysis.” Re-pharase e.g. by adding “it was revealed that” after analysis.

Reply: Have Added“it was revealed that”after analysis. Through conservative domain analysis, it was revealed that all plant PL proteins contain a Pec_Lyase_C domain.

“Members of the Pectin Lyase gene family have made important discoveries …”; it is not the pectin lyases that made important discoveries.

Reply: It has been corrected to “Researchers have made significant discoveries regarding the involvement of pectate lyase gene family members in plant growth and development, fruit softening, and disease resistance.”

“A total of 93 candidate genes with pectin lyase annotations were obtained through a search” be more specific; briefly mention what kind of search, even though you state more details in the Methods section.

Reply: A total of 93 candidate genes with ‘pectate lyase’ annotations were obtained through the terms search in SpudDB database. Added search by term.

“Among the 27 StPL gene family members, 110 were acidic amino acids,” I don’t understand what this means. How can gene family members be amino acids? “while the remaining were neutral or basic amino acids” Same question as above.

Reply: Thank you. This sentence has been changed to “ten out of the 27 StPL gene family members encode acidic amino acids, while the remaining members encode neutral or basic amino acids”.

“and all StPL family members consisted of hydrophilic amino acids” consisted is a strong word; there were surely also some non-polar amino acids? How about “consisted mainly of”? Or “contained mostly”?

Reply: Thank you for your question. A higher number of polar amino acid residues in a protein makes the protein highly hydrophilic, with a negative hydrophilicity index. According to Expasy ProtParam tool, analyzing the protein sequences showed that the Grand Average of Hydropathicity of all StPLs was negative, therefore, the analysis concluded that all StPLs were hydrophilic proteins.

“proteins encoded by genes in this family may be hydrophobic.” Hydrophobic proteins that consist of hydrophilic amino acids? Clarify

Reply: We didn't check it clearly during the writing process, thank you very much for your correction and we apologize for that! We have made revised in the manuscript.

Therefore, it is hypothesized that the proteins encoded by genes in this family may be hydrophilic.

“Figure 6. Cis-acting elements of the potato StPL family, different colors represent different cis-acting elements.” Well, these are “putative” cis-acting elements; they may or may not be significant. If possible, include any confidence estimate, such as E or P values.

Reply: Thanks for the advice. The cis-acting element has no confidence estimate and has been added in the Materials and Methods section to screen motif for E-values < 0.0001.

“We analyzed the expression of each gene in the transcriptome data” mention here how you generated the transcriptome data; cite previous study if appropriate.

Reply: Thank you very much for your correction, the source of material, the method of salt treatment and data analysis have been added in the Materials and Methods section

Analysis of the expression pattern of StPL family genes in potato seedlings under single and mixed salts

The plant material used in this study is diploid potato 320-02. This material is closely related to the reported diploid potato RH, both belonging to Tuberrosum subspecies of potato.320-02 were subjected to different time-length NaCl (100 mmol/L) single salt and equal volume mixed salt treatments of NaCl+Na2SO4 (100 mmol/L), respectively, processing time points were 0, 24, and 48 hours, with 3 replicates per time point. Sampling for transcriptome sequencing.

The resulting transcriptome data were compared to the potato reference genome DM (V6.1) using HISAT2 to construct an index. HISAT2 compared the reads of each sample to the genome to obtain SAM files, which were then sorted and converted to BAM files using Samtools HISAT: a fast spliced aligner with low memory requirements. Nature Methods.Splicing transcripts using StringTie and conducting a reference transcriptome analysis to estimate transcript expression, gene expression, and generate statistics that Ballgown can utilize for further analysis Ballgown bridges the gap between transcriptome assembly and expression analysis. StringTie enables improved reconstruction of a transcriptome from RNA-seq reads. Heatmap of gene expression of pectin lyase gene family using TBtools.

“Figure 7.” What is the unit of the legend? What is “0 to 1” Is it supposed to be log2 fold change, or anything else? State this clearly in the figure.

Reply: Thank you for your correction. Added figure note to figure 7.

The color bar in the figure represents the log2 expression values, and the expression values were normalized using TPM+1.

Figure 8. this could be moved to supplemental material.

Reply: Figure 8 and its annotations have been moved in the supplement material.

Figure 9. “Under confocal microscopy, the cell signal was localized to the membrane system, leading to speculation that StPL18 might be situated in the endoplasmic reticulum (Figure 9e).” It does not look like ER to me; not sure what it is, Golgi? Compare to published dGFP images, or use some known markers yourself for comparison. What does the signal peptide indicate? Using e.g. DeepLoc2.0 could help sorting this out.

Also, describe the experiment in more detail, what is the tissue and plant we are looking at?

Reply: Thank you very much for your valuable suggestions! We replaced the normal subcellular localization results with co-localized subcellular results labeled with endoplasmic reticulum signaling proteins and described the experimental materials and steps in detail in the Materials and Methods section.

To obtain the subcellular localization information of StPL proteins, a fusion protein expression vector of StPL18 and GFP was constructed. GFP was utilized as a reporter gene, and the position of GFP was observed to determine the location of StPL proteins. The recombinant protein, with GFP fused to the coding region of the StPL gene and the stop codon removed, was expressed using the 35S promoter. The subcellular localization of the protein was analyzed later. Sper as an endoplasmic reticulum localization signaling protein to construct a fusion protein with the red fluorescent protein mKATE.

Experimental Steps

(1) Pick Agrobacterium containing the corresponding vectors and single colonies of Agrobacterium containing the P19 plasmid. Inoculate them into 10 mL of LB liquid medium with the corresponding resistance and incubate them for 24 hours at 28°C and 220 rpm until the OD600 value exceeds.

(2) Centrifuge at 5,000 rpm for 2 minutes and discard the liquid medium.

(3) Resuspend the organisms by adding 700 μL of tobacco infestation buffer, mix well, and take 200 μL of the bacterial solution to measure the OD600 value.

(4) Mix Agrobacterium with different vectors in a 2 mL centrifuge tube so that the OD600 value of Agrobacterium containing the P19 plasmid is adjusted to 0.3, and other Agrobacterium is adjusted to 0.5. Leave the mixture at room temperature for about 3 hours. Afterward, carefully pipette the well-grown Agrobacterium from the center of the centrifuge tube and discard the rest.

(5) Take the tobacco leaves (tobacco has about 4 true leaves and is 10 cm tall) with good growth conditions and inject Agrobacterium from the back of the leaves using a 1 mL syringe until the leaves are completely wet.

(6) Incubate in the dark for 24 hours and then transfer to dim light culture for 36 hours.

(7) For co-localization, the marker plasmid was transformed into Agrobacterium, operated in suspension with the constructed vector plasmid Agrobacterium, and mixed at  equivalent ratio before injection into tobacco leaves.

“consist of hydrophilic amino acids, which are chemically unstable.” The amino acids are unstable? I don’t understand this.

Reply: We have added information in Table 1 about whether an amino acid is stable or not, which is based on the instability index. Then it is judged by its polarity based on the hydrophilic index. We made an error in the previous judgment and wrote the words hydrophilic and hydrophobic incorrectly, so that caused your confusion. We have rewritten the word hydrophilic and hydrophobic, so it caused your confusion. We have rewritten it, thanks for your correction!

“It was speculated that StPL18 might act as an activator of specific defense responses, enhancing the sensitivity of potatoes to salt stress.” Where was this speculated, add citation. Also, why would activation of defense enhance the sensitivity to salt stress? Explain.

Reply: Thanks for the correction, we meant that the upregulation of the expression of this gene increased the plant's sensitivity to salt, which has been corrected in the manuscript.

“The subcellular localization of StPL18 indicates that it may be situated in the endoplasmic reticulum.” Again, I’m not convinced of this.

Reply: Thanks to your correction, we replaced the results of subcellular localization to subcellular co-localization results, according to which the target gene localization position in tobacco leaves overlapped with the endoplasmic reticulum marker position, and there were more obvious cytoplasmic granular aggregates.

Materials and Methods

“Search for the keywords "Pectate lyase"” but the paper was on “pectin lyases. Which is it?

Reply: Both terms refer to pectate lyase enzyme (PL, EC 4.2.2.2), for which different researchers have different preferences and which have been harmonized in the manuscript.

“Potato materials were subjected to different time-length NaCl (100 mmol/L) single salt and equal volume mixed salt treatments of NaCl+Na2SO4 (100 mmol/L), respectively, to analyze the expression of each gene member of the family in the transcriptome data”. I don’t understand how the transcriptome data was generated. After the plants were subjected to salt, what was done next? Or is this data-mining? In that case, don’t present it like plants were actually treated in this study.

Reply: Thank you for the correction, we have supplemented the Materials and Methods section with sources of transcriptomic data and methods of analysis. Currently, only the genes of this family have been analyzed for different salt stresses.

Reviewer 2 Report

Comments and Suggestions for Authors

The authors have identified the 27 members of the potato pectin lyase gene family through a series of bioinformatics analyses. These members were categorized into eight groups using phylogenetic analyses. The promoters of some members contained hormone- and stress-responsive elements, and some responded positively to salt treatment. This research lays the groundwork for further study on lyase enzymes. The reviewer appreciates the effort of the authors to prove their hypothesis using bioinformatics tools. However, the reviewer has a few comments regarding this study. Thus, the authors need to consider the following comments to improve the quality of this manuscript.

Authors are advised to write two lines about the background of the study in abstract.

Please improve the abstract more with study results.

Introduction last paragraph should be revised with objectives and rationale of the study.

Authors are advised to fix grammar, space, punctuation, and formatting errors throughout the manuscript.

Provide space between manuscript sentences and reference numbers.

Plant scientific names should be in full form in first mention rest should be abbreviated. Authors should revise this throughout the manuscript. Eg. Line 67 - Arabidopsis thaliana should be A. thaliana. Follow the same throughout the manuscript.

Line 66, 67, 149, 154, 160, etc.: Scientific name should be italics. Check and revise in the entire manuscript.

Line 82, 95: Use ‘this present study/current study’ instead of ‘this paper’ follow the same in the entire manuscript.

All figure legends should be improved with clear details instead of just mentioning the caption. It should be self-explanatory. Please refer some related papers for reference.

Line 129: Syntax error. Reframe it. “Vary widely and are widely”.

Add amino acid length and stability/unstable information to the Table 1.

In section 2.5: Authors have only mentioned the 27 PL genes divided into 8 groups. On what basis it was categorized? What each group represents like domain or function difference needs to be mentioned. Please refer to some related papers.

The results section needs to be improved, well with details. It seems the authors have hastily written the manuscript.

The discussion section looks shallow, improve it with appropriate reference and details.

Why authors did not performed the real-time PCR for understanding the PL genes expression regulation under salt stress conditions?

If possible, authors are advised to perform gene co-expression analysis.

Authors must concentrate on the formatting and use of symbols, etc., throughout the manuscript.

In conclusion section and write a few lines about future perspectives or hypotheses about the study. It will be useful to the readers for ease of understanding and carry out further research.

Comments on the Quality of English Language

Please refer the comments

Author Response

The authors have identified the 27 members of the potato pectin lyase gene family through a series of bioinformatics analyses. These members were categorized into eight groups using phylogenetic analyses. The promoters of some members contained hormone- and stress-responsive elements, and some responded positively to salt treatment. This research lays the groundwork for further study on lyase enzymes. The reviewer appreciates the effort of the authors to prove their hypothesis using bioinformatics tools. However, the reviewer has a few comments regarding this study. Thus, the authors need to consider the following comments to improve the quality of this manuscript.

Authors are advised to write two lines about the background of the study in abstract.

Reply: Thanks for the suggestion, we have added some research background in the abstract section.

Pectin is a structural polysaccharide and a major component of plant cell walls. Pectate lyases, are a class of enzymes that degrade demethylated pectin by cleaving the α-1,4-glycosidic bond, and they play an important role in plant growth and development. Currently, little is known about the PL gene family members and their involvement in salt stress in potato.

Please improve the abstract more with study results.

Reply: Thanks for the suggestion. The results of subcellular localization have been added to the summary

Since StPL18 exhibited a consistent expression pattern under both single and mixed salt stress conditions, its subcellular localization was determined. The results indicated that StPL18 is localized in the endoplasmic reticulum membrane.

Introduction last paragraph should be revised with objectives and rationale of the study.

Reply: Thank you for the suggestion. We rewrote the last paragraph of the introduction to make it more details.

In summary, this research is based on the DM v6.1 Genome assembly for the doubled monoploid potato information. It identifies and analyzes the members of the StPL gene family, while also examining the expression of these genes under salt stress. These findings provide a basis for further analysis of the differentiation process and biological function of the StPL gene family in potato. These results laid the foundation for further analyzing the differentiation process and biological functions of the StPL gene family in potato and provided a theoretical basis for the discovery and screening of stress resistance genes in potato.

Authors are advised to fix grammar, space, punctuation, and formatting errors throughout the manuscript.

Reply: Thanks for the suggestion, we apologize for the formatting errors, we've checked them throughout and corrected them, hopefully they won't show up again!

Provide space between manuscript sentences and reference numbers.

Reply: Spaces have been added before all references numbers. Thanks for your reminder.

Plant scientific names should be in full form in first mention rest should be abbreviated. Authors should revise this throughout the manuscript. Eg. Line 67 - Arabidopsis thaliana should be A. thaliana. Follow the same throughout the manuscript.

Reply: Thank you for the suggestion. Full text have checked and revised.

Line 66, 67, 149, 154, 160, etc.: Scientific name should be italics. Check and revise in the entire manuscript.

Reply: We apologize for these minor errors and thank you for taking the time to correct us! Throughout the manuscript, we have checked and revised.

Line 82, 95: Use ‘this present study/current study’ instead of ‘this paper’ follow the same in the entire manuscript.

Reply:Thank you for your correction. We have replaced ‘this paper’ with ‘this study’.

All figure legends should be improved with clear details instead of just mentioning the caption. It should be self-explanatory. Please refer some related papers for reference.

Reply: Thanks for the correction. We have referred to other papers to provide figure notes with more details.

Line 129: Syntax error. Reframe it. “Vary widely and are widely”.

Reply: Thanks for the correction. Have reframed the sentence.

The above results indicate that the molecular weights of different genes varied and were widely distributed.

Add amino acid length and stability/unstable information to the Table 1.

Reply: Thank you for your suggestion, we have added amino acid length information in column 4 and stability information in column 7 of the table, as well as the transcript ID of the gene in the second column, which we believe will make the table easier to interpret.

In section 2.5: Authors have only mentioned the 27 PL genes divided into 8 groups. On what basis it was categorized? What each group represents like domain or function difference needs to be mentioned. Please refer to some related papers.

Reply: Thank you for your correction. We introduced homologous genes from Arabidopsis thaliana, tomato and rice into the evolutionary tree, and therefore referred to their classification of PL genes to make the classification results of StPL. Citations have been added at the beginning of the paragraphs and the field or functional differences represented by each group have been mentioned.

The results section needs to be improved, well with details. It seems the authors have hastily written the manuscript.

Reply: We apologize for this, thank you for the correction, and the results section has been checked and improved throughout!

The discussion section looks shallow, improve it with appropriate reference and details.

Reply: Thanks again for the correction, we have reorganized the discussion section to make it more logical and revised some of the references.

Why authors did not performed the real-time PCR for understanding the PL genes expression regulation under salt stress conditions?

Reply: Thank you for your question. We will verify the salt response of PL genes by RT-PCR at a later and conduct functional analysis of their salt tolerance and molecular mechanisms.

If possible, authors are advised to perform gene co-expression analysis.

Reply: Thank you for your suggestion, we will do further research at a later stage, when we will conduct co-expression analysis, thank you again for your guidance!

Authors must concentrate on the formatting and use of symbols, etc., throughout the manuscript.

Reply: Thanks for the correction, it's been checked throughout.

In conclusion section and write a few lines about future perspectives or hypotheses about the study. It will be useful to the readers for ease of understanding and carry out further research.

Reply: Thanks to your suggestion, we have added the future perspectives in the conclusion section.

In this study, 27 StPL genes in potato were screened using salt treatments. Eventually, the StPL18 protein, located in the endoplasmic reticulum, garnered attention. It was discovered to regulate the morphogenesis of potato, as previously researched and StPL18 was up-regulated under different salt treatments.  Elevated expression of pectin cleavage enzymes destabilizes the cell wall structure, compromising its protective function. This allows harmful ions like Na+ to penetrate the cell, increasing the plant's salt sensitivity. This sensitivity can be heightened by acquiring material from the pl mutant. Mutations in the PL gene impact the weight of potato tubers but result in more branches and tubers, which is advantageous for certain consumers preferring smaller tubers. Therefore, this gene may hold significant value in enhancing potato salt tolerance and cultivating new varieties.

Round 2

Reviewer 1 Report

Comments and Suggestions for Authors

Most of my suggestions have been incorporated in the revised manuscript. I just have a few minor comments (listed below); once these are addressed, I believe the manuscript is ready for publication.

“Among the 27 StPL gene family members, 110 were acidic amino acids,” I don’t understand what this means. How can gene family members be amino acids? “while the remaining were neutral or basic amino acids” Same question as above. 

Reply: Thank you. This sentence has been changed to “ten out of the 27 StPL gene family members encode acidic amino acids, while the remaining members encode neutral or basic amino acids”.

REVIEWER: This is still not correct. This makes it sound like “ten out of the 27 StPL proteins consist entirely of acidic amino acids, which is of course not the case. Better state that these genes encode acidic proteins, while the remaining members encode neutral or basic proteins.

“and all StPL family members consisted of hydrophilic amino acids” consisted is a strong word; there were surely also some non-polar amino acids? How about “consisted mainly of”? Or “contained mostly”? 

Reply: Thank you for your question. A higher number of polar amino acid residues in a protein makes the protein highly hydrophilic, with a negative hydrophilicity index. According to Expasy ProtParam tool, analyzing the protein sequences showed that the Grand Average of Hydropathicity of all StPLs was negative, therefore, the analysis concluded that all StPLs were hydrophilic proteins.

REVIEWER: Yes, but hydrophilic proteins do not consist entirely of hydrophilic amino acids. You can’t state that “all StPL family members consisted of hydrophilic amino acids”, which makes it sound like hydrophilic amino acids are the only type of amino acids found in these proteins. 

“consist of hydrophilic amino acids, which are chemically unstable.” The amino acids are unstable? I don’t understand this. 

Reply: We have added information in Table 1 about whether an amino acid is stable or not, which is based on the instability index. Then it is judged by its polarity based on the hydrophilic index. We made an error in the previous judgment and wrote the words hydrophilic and hydrophobic incorrectly, so that caused your confusion. We have rewritten the word hydrophilic and hydrophobic, so it caused your confusion. We have rewritten it, thanks for your correction!

REVIEWER: I don’t think the instability index “is judged by its polarity based on the hydrophilic index”. Protparam states: “The instability index provides an estimate of the stability of your protein in a test tube. Statistical analysis of 12 unstable and 32 stable proteins has revealed [7] that there are certain dipeptides, the occurence of which is significantly different in the unstable proteins compared with those in the stable ones.” 

Author Response

Most of my suggestions have been incorporated in the revised manuscript. I just have a few minor comments (listed below); once these are addressed, I believe the manuscript is ready for publication.

“Among the 27 StPL gene family members, 110 were acidic amino acids,” I don’t understand what this means. How can gene family members be amino acids? “while the remaining were neutral or basic amino acids” Same question as above.

Reply: Thank you. This sentence has been changed to “ten out of the 27 StPL gene family members encode acidic amino acids, while the remaining members encode neutral or basic amino acids”.

REVIEWER: This is still not correct. This makes it sound like “ten out of the 27 StPL proteins consist entirely of acidic amino acids, which is of course not the case. Better state that these genes encode acidic proteins, while the remaining members encode neutral or basic proteins.

Reply: Thank you for taking the trouble to correct me. The sentence was modified to ‘Ten out of the 27 members encode acidic proteins, while the remaining members encode neutral or basic proteins’.

“and all StPL family members consisted of hydrophilic amino acids” consisted is a strong word; there were surely also some non-polar amino acids? How about “consisted mainly of”? Or “contained mostly”?

Reply: Thank you for your question. A higher number of polar amino acid residues in a protein makes the protein highly hydrophilic, with a negative hydrophilicity index. According to Expasy ProtParam tool, analyzing the protein sequences showed that the Grand Average of Hydropathicity of all StPLs was negative, therefore, the analysis concluded that all StPLs were hydrophilic proteins.

REVIEWER: Yes, but hydrophilic proteins do not consist entirely of hydrophilic amino acids. You can’t state that “all StPL family members consisted of hydrophilic amino acids”, which makes it sound like hydrophilic amino acids are the only type of amino acids found in these proteins.

Reply: Thank you for your advice. For once, I get your point. The sentence was modified to ‘The analysis based on the predicted values of the grand average of hydropathicity showed that all the values were negative, and the aliphatic index varied from 68.52 to 92.89, indicating that most of these proteins exhibit some level of hydrophilicity’.

“consist of hydrophilic amino acids, which are chemically unstable.” The amino acids are unstable? I don’t understand this.

Reply: We have added information in Table 1 about whether an amino acid is stable or not, which is based on the instability index. Then it is judged by its polarity based on the hydrophilic index. We made an error in the previous judgment and wrote the words hydrophilic and hydrophobic incorrectly, so that caused your confusion. We have rewritten the word hydrophilic and hydrophobic, so it caused your confusion. We have rewritten it, thanks for your correction!

REVIEWER: I don’t think the instability index “is judged by its polarity based on the hydrophilic index”. Protparam states: “The instability index provides an estimate of the stability of your protein in a test tube. Statistical analysis of 12 unstable and 32 stable proteins has revealed [7] that there are certain dipeptides, the occurence of which is significantly different in the unstable proteins compared with those in the stable ones.”

Reply: Thank you very much for your careful guidance, we have further revised this part of the description.

The analysis based on the predicted values of the grand average of hydropathicity showed that all the values were negative, and the aliphatic index varied from 68.52 to 92.89, indicating that most of these proteins exhibit some level of hydrophilicity.

Reviewer 2 Report

Comments and Suggestions for Authors

The authors have suitably incorporated all my suggestions into the revised manuscript. Now the manuscript has been improved more than before. Therefore, I recommend this manuscript be accepted for publication.

Author Response

Thank you very much for your proposed revisions to our manuscript!